# An oxytocin/vasopressin-related neuropeptide modulates social foraging behavior in the clonal raider ant

Ingrid Fetter-Pruneda [1,2]*, Taylor Hart [1], Yuko Ulrich[1,3], Asaf Gal [1], Peter R. Oxley [1,4], Leonora Olivos-Cisneros[1], Margaret S. Ebert[5], Manija A. Kazmi[6], Jennifer L. Garrison [5,7], Cornelia I. Bargmann [5,8], Daniel J. C. Kronauer [1]*

**1** Laboratory of Social Evolution and Behavior, The Rockefeller University, New York, New York, United States of America, **2** Departamento de Biología Celular y Fisiología, Instituto de Investigaciones Biomédicas, Universidad Nacional Autónoma de México, Mexico City, Mexico, **3** Institute for Integrative Biology, ETH Zurich, Zurich, Switzerland, **4** Samuel J. Wood Library, Weill Cornell Medicine, New York, New York, United States of America, **5** Lulu and Anthony Wang Laboratory of Neural Circuits and Behavior, The Rockefeller University, New York, New York, United States of America, **6** Laboratory of Chemical Biology and Signal Transduction, The Rockefeller University, New York, New York, United States of America, **7** Buck Institute for Research on Aging, Novato, California, United States of America, **8** Chan Zuckerberg Initiative, Redwood City, California, United States of America

\* ifetter@iibiomedicas.unam.mx (IFP); dkronauer@rockefeller.edu (DJCK)

**Data Availability Statement:** Relevant data are within the paper and its Supporting Information files. Further supporting data can be found at: https://github.com/oxpeter/ooceraea_inotocin_

## Abstract

Oxytocin/vasopressin-related neuropeptides are highly conserved and play major roles in regulating social behavior across vertebrates. However, whether their insect orthologue, inotocin, regulates the behavior of social groups remains unknown. Here, we show that in the clonal raider ant *Ooceraea biroi*, individuals that perform tasks outside the nest have higher levels of inotocin in their brains than individuals of the same age that remain inside the nest. We also show that older ants, which spend more time outside the nest, have higher inotocin levels than younger ants. Inotocin thus correlates with the propensity to perform tasks outside the nest. Additionally, increasing inotocin pharmacologically increases the tendency of ants to leave the nest. However, this effect is contingent on age and social context. Pharmacologically treated older ants have a higher propensity to leave the nest only in the presence of larvae, whereas younger ants seem to do so only in the presence of pupae. Our results suggest that inotocin signaling plays an important role in modulating behaviors that correlate with age, such as social foraging, possibly by modulating behavioral response thresholds to specific social cues. Inotocin signaling thereby likely contributes to behavioral individuality and division of labor in ant societies.

## Introduction

Oxytocin/vasopressin-related neuropeptides are highly conserved and play major roles in regulating social behavior in vertebrates [1–10], and animals from jellyfish to humans possess orthologues. In mammals, oxytocin and vasopressin regulate diverse social behaviors such as parental care, pair bonding, social cognition, and aggression [1–7]. However, their role in

expression (DOI: 10.5281/zenodo.4562942) and https://doi.org/10.5281/zenodo.4813078.

**Funding:** Research reported in this publication was supported by the National Institute of General Medical Sciences of the National Institutes of Health under Award Number R35GM127007 to D. J.C.K. The content is solely the responsibility of the authors and does not necessarily represent the official views of the National Institutes of Health. This work was also supported by a Sinsheimer Scholar Award, a Klingenstein-Simons Fellowship Award in the Neurosciences, and a Faculty Scholars Award from the Howard Hughes Medical Institute to D.J.C.K., a G. Harold & Leila Y. Mathers Charitable Foundation award to C.I.B., by the Howard Hughes Medical Institute, of which C.I.B. was an Investigator, and Fulbright-García Robles and Rockefeller University Women & Science fellowships to I.F.-P. The funders had no role in study design, data collection and analysis, decision to publish, or preparation of the manuscript.

**Competing interests:** The authors have declared that no competing interests exist.

**Abbreviations:** AMMC, antennal mechanosensory and motor center; BSA, bovine serum albumin; GPCR, G protein–coupled receptor; HCR, hybridization chain reaction; HEK293T, human embryonic kidney 293T; PKG, cGMP-dependent protein kinase; r.m.s.d., root–mean–square deviation; RNAi, RNA interference; RNA-seq, RNA sequencing; SEZ, subesophageal zone; sNPFR, short neuropeptide F receptor.

regulating social behavior in insects remains less well understood [11], in part because they are absent in prominent model organisms like fruit flies and honeybees. Ants, on the other hand, have an oxytocin/vasopressin-related orthologue called inotocin [11,12], providing a unique opportunity to study its potential role in modulating social behavior and division of labor in highly social insects. Division of labor is the hallmark of insect societies and is thought to form the basis of their ecological dominance [13]. In ants and bees, queens lay eggs, and workers perform tasks such as nursing, foraging, nest maintenance, and defense [14,15]. Division of labor is associated with age, genotype, individual experience, size, and morphology [16,17] and is characterized by both specialization and behavioral flexibility [16]. Workers engaged in a given task can switch tasks in response to changes in colony needs and social context. Which individuals in a colony forage is thought to be determined by differences in behavioral response thresholds to social signals and environmental cues [16,18–23]. For example, individuals that are more responsive to hungry larvae might have a higher tendency to leave the nest and forage than less responsive individuals. Such putative differences in behavioral response thresholds could underlie the stable behavioral division of labor observed in insect societies [18,19]. Behavioral task allocation among workers is often a function of age, a phenomenon known as age polyethism. In the societies of ants, bees, wasps, and termites, older workers tend to leave the nest and forage, while younger workers tend to perform tasks inside the nest, suggesting that behavioral response thresholds undergo stereotypical changes as a function of age (e.g., [24–30]). Furthermore, the propensity to leave the nest and forage can also be a function of an individual's reproductive physiology or genotype (e.g., [31–34]).

Biogenic amines such as octopamine [35], serotonin [36], and dopamine [37,38], as well as the neuropeptide corazonin [39], have been shown to regulate behavior in ants. In addition, in ant species with multiple worker subcastes, the *foraging* gene, which encodes a cGMP-dependent protein kinase (PKG), has been shown to underlie behavioral flexibility according to colony needs irrespective of ant morphology [40]. A previous study using RNA interference (RNAi)-mediated knockdown of inotocin expression in *Lasius* ants found that decreased gene expression led to elevated locomotor and self-grooming activity, but had no effect on social behaviors such as trophallaxis, allogrooming, or antennation [41]. It has also been demonstrated that inotocin signaling in ants affects the expression of genes involved in metabolism in general and the synthesis of cuticular hydrocarbons in particular [41,42]. Expression of *inotocin* and its receptor also increases with age, suggesting a role in desiccation resistance in foragers [42]. However, whether inotocin plays a role in regulating ant behavior in a social context is unknown, and the behavioral effect of artificially increasing inotocin peptide levels has not been tested.

We studied this question in the clonal raider ant *Ooceraea biroi*, an emerging model system for the study of complex social behavior. Colonies of *O. biroi* consist of totipotent workers that reproduce synchronously and clonally [43–47]. Colonies alternate between a reproductive phase during which the workers lay eggs and a brood care phase during which workers display division of labor with some ants foraging and others nursing the larvae [46,48,49]. This cycle is regulated by the presence of larvae, which suppress reproduction and induce brood care behavior (i.e., foraging and nursing) in the adults [46,48–50]. Moreover, as in other social insects, the propensity to forage correlates with age, with older ants performing more tasks outside the nest than younger ants (age polyethism) [51]. Finally, workers have a variable number of ovarioles, which correlates with behavior, with regular workers (2 or 3 ovarioles) performing more foraging activity than intercastes (4 or more ovarioles) [51]. Moreover, in *O. biroi*, stable division of labor emerges even among workers that are matched for genotype, age, and reproductive physiology, suggesting underlying variation in behavioral response thresholds [20,51].

## Results and discussion

The insect inotocin peptide is an orthologue of the neuropeptides oxytocin and vasopressin in humans and other mammals [11,12] (Fig 1A). The *inotocin* gene codes for the inotocin pre-pro-peptide in ants. We used reciprocal BLAST searches to identify this gene in the genome of the clonal raider ant *O. biroi* [47]. Oxytocin and vasopressin signal through G protein–coupled receptors (GPCRs) and bioinformatics analyses of ant genomes predict one receptor with sequence homology to mammalian oxytocin and vasopressin receptors [12]. We extracted RNA from *O. biroi* and sequenced full-length cDNA clones of the putative inotocin receptor, identifying 10 alternative splice variants that were also supported by genome-wide RNA sequencing (RNA-seq) data [52] (Fig 1B, S1 Fig). Of the 10 splice variants, only 2 encode all 7 transmembrane domains characteristic of GPCRs. Based on the structural information available for the human oxytocin receptor [53,54], only 2 of these 10 isoforms likely produce functional receptors that can be trafficked to the cell membrane and bind inotocin. While the significance of these alternative splice variants remains unknown, our finding suggests that splicing of the inotocin receptor mRNA is dynamic and possibly important for regulating inotocin signaling. For example, alternative splicing could regulate receptor expression, or the shorter proteins could interfere with trafficking or function of full-length isoforms [55]. We then cloned and expressed the putative full-length *O.biroi* inotocin receptor cDNA in HEK293T cells and exposed cells to a synthetic version of inotocin. Cells transfected with the inotocin receptor responded with calcium signals at a picomolar half-maximal effective concentration ($EC_{50}$) (Fig 1C, S2 and S3 Figs for experiment controls) and also produced calcium transients when stimulated with human oxytocin and vasopressin peptides (S4 Fig). This demonstrates that the *O. biroi* receptor is functional, matching previous findings from the ants *Lasius niger*, *Lasius neglectus*, and *Camponotus fellah* [41,42,56]. We also cloned one of the splice variants that only codes for the first 3 transmembrane domains. Calcium imaging showed that, as expected, it does not respond to inotocin when expressed in HEK293T cells (S5 Fig).

To localize inotocin-producing cells and their projections in the ant brain, we used a custom antibody that specifically recognizes the inotocin pre-pro-peptide (see Materials and methods for details). Our antibody stained 2 neurons in the subesophageal zone (SEZ) of the *O. biroi* brain, localizing to the cytoplasm and excluded from the nucleus. These 2 neurons project throughout the SEZ and to the antennal mechanosensory and motor center (AMMC), as demonstrated by the presence of punctae stained for inotocin, some of which are likely in projections (Fig 2A–2C). The cell bodies stained by antibody were also labeled by in situ hybridization for the *inotocin* mRNA (Fig 2B and 2D). We did not observe staining with this antibody in *Drosophila melanogaster*, which lacks an oxytocin/vasopressin-related peptide. Together with other negative controls, this indicates that our antibody recognizes inotocin and not unintended targets (S6 Fig).

Two inotocin-producing cells in the brain region analogous to the SEZ have previously been identified in ants of the genera *Lasius* and *Camponotus* in the subfamily Formicinae [41,42]. In concert with our findings in the clonal raider ant, a representative of a second major ant subfamily, the Dorylinae, this suggests that inotocin signaling is indeed a conserved feature of ant biology. Potentially homologous cells have also been found in homologous brain regions in distantly related insects such as beetles [57], cockroaches [58], mantids, grasshoppers, and locusts [59,60]. Unlike in *O. biroi* and *Lasius*, however, a study of *C. fellah* found additional inotocinergic neurons in the protocerebrum [42], implying a significant level of evolutionary lability in inotocin expression patterns across the ants.

**A**

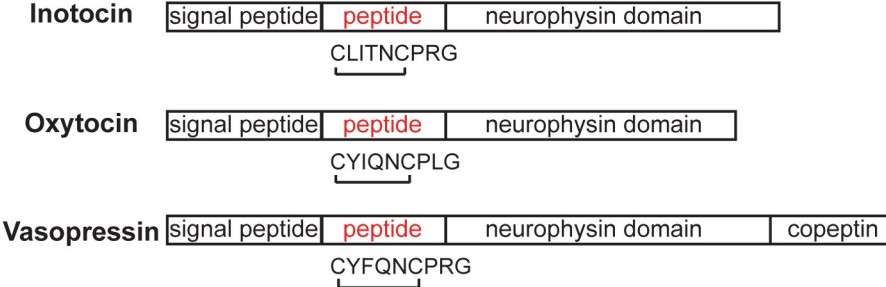

**B**

### Inotocin receptor isoforms

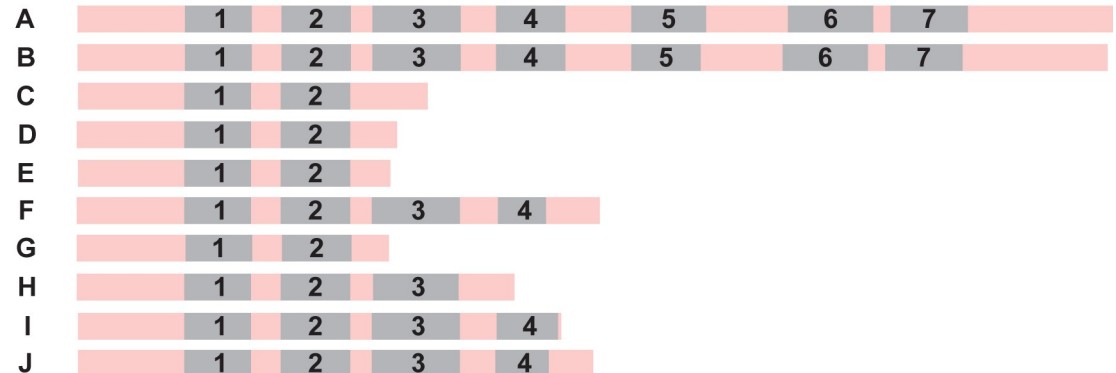

**C**

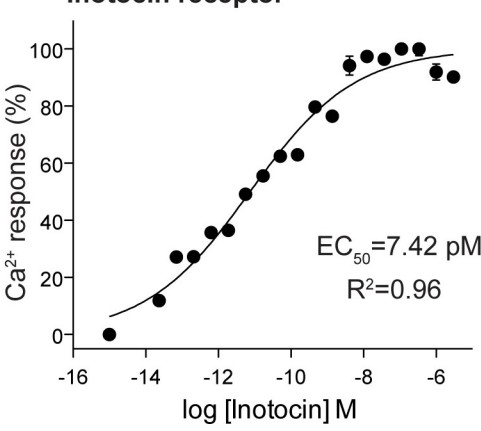

**Fig 1. Inotocin and its receptor in the clonal raider ant. (A)** Pre-pro-peptide domain structure and amino acid sequence of *O. biroi* inotocin, compared to human oxytocin and vasopressin. **(B)** Schematic representation of the 10 different isoforms that the *O. biroi inotocin receptor* gene produces. The transmembrane domains required for receptor function are marked in gray. **(C)** Example of a dose–response curve of FLIPR-calcium 4-loaded HEK293T cells transfected with the *inotocin* receptor and exposed to synthetic inotocin peptide. The $EC_{50}$ was calculated with nonlinear regression using a 4-parameter logistic curve fit. The data underlying this figure can be found in S1 Data. See S2–S5 Figs for controls.

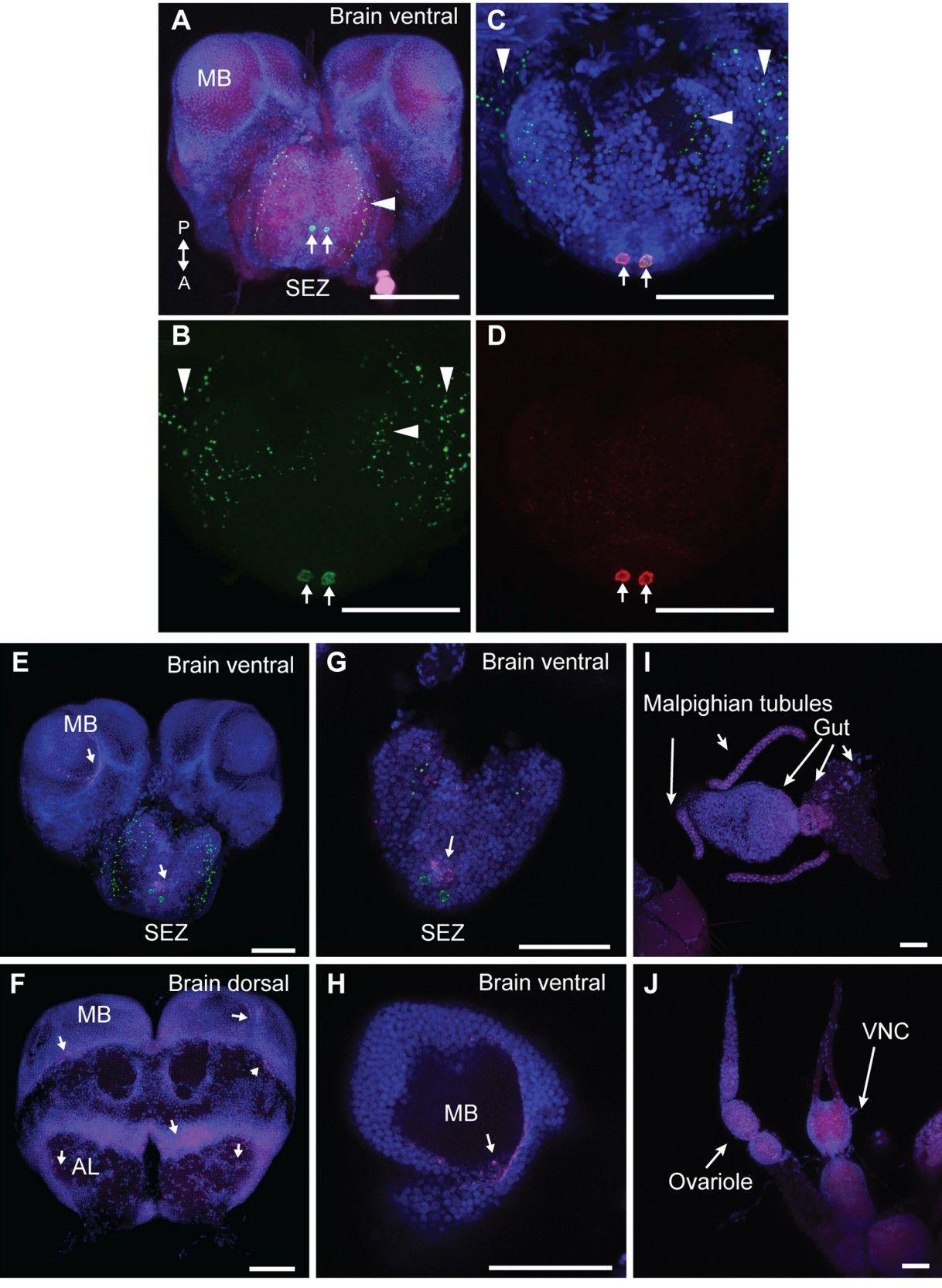

**Fig 2. A single pair of inotocinergic neurons is localized in the SEZ of the clonal raider ant brain. The *inotocin receptor* is expressed in the brain, VNC, ovary, gut, and Malpighian tubules. (A)** Ventral view of the ant brain immunostained for the inotocin pre-pro-peptide (green). **(B)** Magnified ventral view of the ant brain, showing the inotocin immunostain (green) in the SEZ. **(C)** Same as in B, but showing co-localization of inotocin immunostain (green) and in situ hybridization for *inotocin* (red). **(D)** Same as in B, but showing only in situ hybridization for *inotocin* (red). Nuclei are stained with DAPI (blue) in A and C, and actin is stained with phalloidin (magenta) in A. Scale bars represent 100 μm in A and 50 μm in B–D. The 2 inotocinergic neurons in the SEZ are indicated by arrows, and inotocin-stained punctae in the SEZ and AMMC are indicated by arrowheads. **(E–J)** Broad expression of *itcR* in the brain and body as detected through *itcR* in situ hybridization (magenta). (E and F) *itcR* is expressed throughout the brain, including the SEZ and the MBs (arrows). Higher magnification view of the brain shown in A of SEZ (G) and MBs (H). (I) *itcR* is expressed in Malpighian tubules and the gut (arrows). (J) *itcR* is expressed in the VNC and ovarioles (arrows). *itcR* in situ hybridization is shown in magenta, nuclei are stained with DAPI (blue), and anti-inotocin peptide staining is shown in green. Scale bars represent 50 μm in E–J. AL, antennal lobe; AMMC, antennal mechanosensory and motor center; MB, mushroom body; SEZ, subesophageal zone; VNC, ventral nerve cord.

The localization of a neuropeptide in a particular brain region can point to the respective peptide being a modulator of neuronal sensitivity and activity of cells where the neuropeptide's receptor is expressed [61,62]. We found through in situ hybridization that the inotocin receptor is expressed in the SEZ as well as throughout the brain, ventral nerve cord, Malpighian tubules, gut, and ovaries, consistent with a possible role for inotocin in the regulation of both behavior and physiology (Fig 2E–2J). The SEZ functions as the taste and feeding center of the insect brain [63]. Notably, studies in mammals suggest that oxytocin and vasopressin neurons evolved from an ancient cell type with sensory and neurosecretory properties that integrate reproductive functions with energy status and feeding behavior [64]. In accordance with this idea, oxytocin has been linked to the regulation of foraging behavior in mammals [65–67], and *inotocin* transcription levels in *Lasius* ants change in response to starvation and feeding [41]. We therefore hypothesized that inotocin plays a role in modulating the neuronal circuitry underlying foraging behavior in ants. Unlike in solitary organisms, foraging in ants and honeybees is a social behavior that is regulated by the colony environment. In *O. biroi*, for example, workers forage in response to hungry larvae in a dose-dependent manner [50].

To investigate whether the inotocin peptide plays a role in modulating behavioral response thresholds, we used automated behavioral tracking of *O. biroi* workers during the brood care phase in the presence of larvae and recorded their behavioral propensities (S7 Fig). Once the larvae had pupated and foraging activity had ceased, we stopped the recordings and analyzed the ants' behavior to identify specialized "nurses" (ants that mostly remained inside the nest) and specialized "foragers" (ants that frequently left the nest). We then dissected the brains of these identified ants at the beginning of the next brood care phase (S7 Fig; individual behavioral tendencies are stable across subsequent colony cycles [20]). Unlike previous studies of ant inotocin that measured RNA levels, we directly assayed peptide levels by measuring antibody staining intensities (see Materials and methods). We found that age-matched (1 month old) ants that had spent more time outside the nest during the previous brood care phase (foragers) had significantly higher levels of inotocin in the cell bodies than ants that had mostly stayed inside the nest (nurses) ($p = 0.022$; Fig 3A). These differences were moderate, however, and no difference was detectable at the level of inotocin peptide punctae (S8 Fig). We then asked whether inotocin levels change as a function of age. We found that older ants (approximately 20 weeks of age) had considerably more inotocin in their cell bodies than young ants (approximately 3 weeks of age) ($p = 0.0002$; Fig 3B). The same was true for inotocin punctae ($p = 0.045$ for number of inotocin punctae; $p < 0.0003$ for fluorescence intensity in inotocin punctae; S8 Fig). As in other social insects, older *O. biroi* workers forage more than younger workers [17]. Thus, across 2 contexts, a higher propensity to forage correlated with higher inotocin peptide levels. These findings are consistent with previous studies on formicine ants, showing that *inotocin* gene expression is elevated in foragers compared to nurses and older compared to younger workers [41,42]. Finally, the propensity to forage also correlates with reproductive

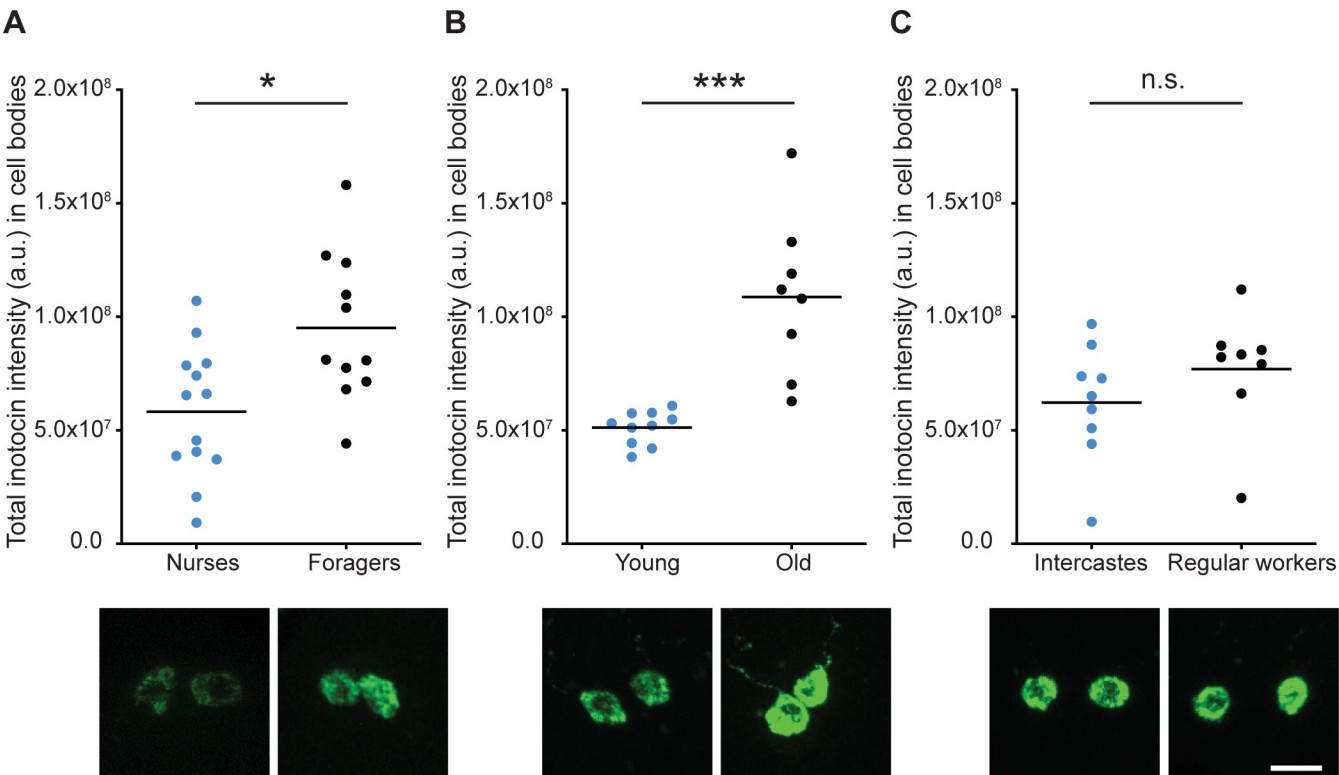

**Fig 3. Foragers and old ants have higher levels of inotocin in the brain. (A)** Among aged-matched ants, foragers have more inotocin than nurses inside inotocinergic cell bodies ($n \geq 11$; unpaired $t$ test, $^*p = 0.022$, Bonferroni corrected; measured during the brood care phase). **(B)** Old ants have more inotocin in cell bodies than young ants ($n \geq 8$; unpaired $t$ test, $^{***}p = 0.0003$, Bonferroni corrected; measured during the brood care phase). **(C)** No difference in inotocin levels was observed between intercastes and age-matched regular workers ($n \geq 8$; unpaired $t$ test, $p = 0.785$, Bonferroni corrected; measured during the brood care phase). Levels of inotocin were calculated from anti-inotocin (neurophysin domain) stained brains by measuring the total intensity of all fluorescent voxels from each image stack. Representative cell body confocal images are displayed under each experimental condition labeled in the graphs. The scale bar represents 10 μm. The data underlying this figure can be found in S2 Data.

physiology in *O. biroi*, with ants that have fewer ovarioles being more active outside the nest than ants with more ovarioles [45]. Ants with more ovarioles are also less suppressed in their reproductive activity by larval signals [48,68]. However, we did not detect any differences in inotocin levels between ants with different reproductive physiologies (Fig 3C, S8 Fig). Taken together, these results open the possibility that inotocin plays a role in modulating foraging propensity in ants, and, specifically, that it might underlie aspects of age polyethism. At the same time, behavioral differences between workers with different reproductive physiologies seem to be regulated by other mechanisms.

To investigate whether inotocin is directly involved in regulating social behavior and division of labor, we used pharmacological manipulations to examine the potential causality between inotocin levels and foraging propensity. First, we established the capacity to modulate the behavior of ants using a whole body immersion pharmacological treatment, by comparing the change in behavior of ant colonies (containing a mix of young and old ants; see Materials and methods) treated with inotocin versus colonies treated with a scrambled control peptide. We found that, over a 5-hour time course following treatment, ants treated with inotocin spent more time outside the nest than controls, suggesting that inotocin can increase the propensity to perform tasks outside the nest ($p = 0.0002$; S9 Fig).

We then performed a second experiment to study the effect of the social context and age on the action of inotocin, in which the age structure and composition of experimental colonies

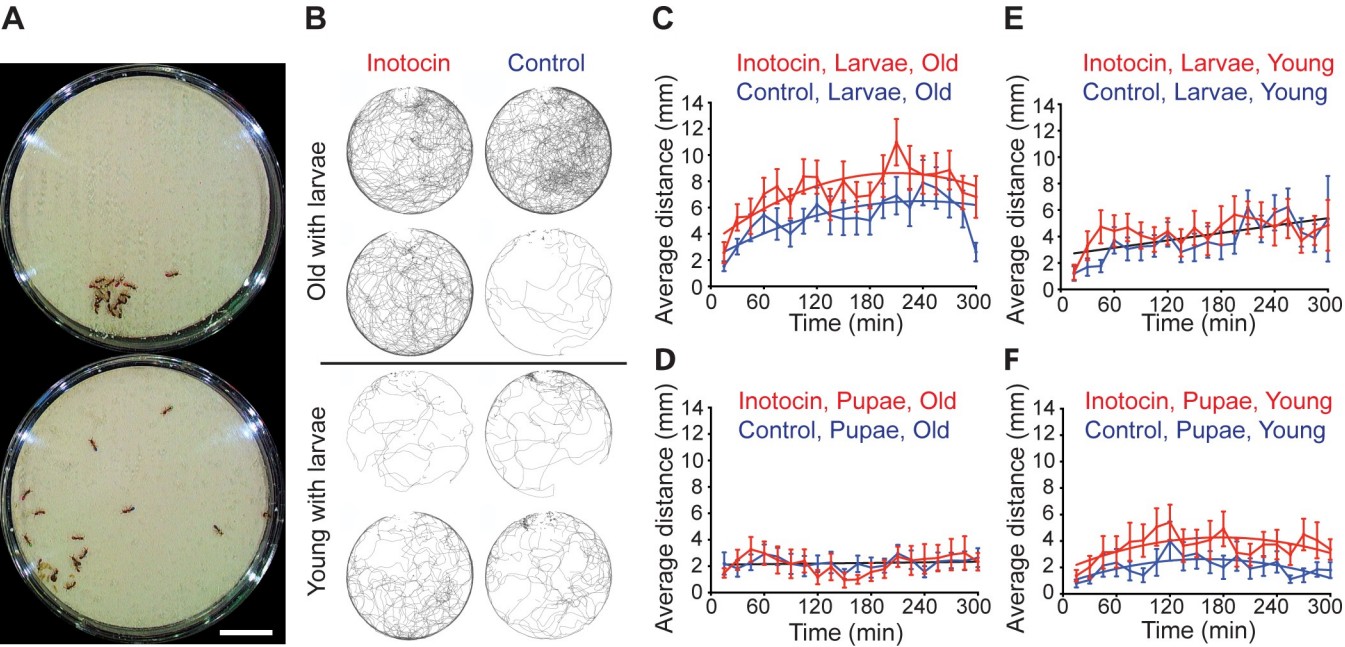

**Fig 4. Elevated inotocin levels lead to increased foraging in an age- and social context–dependent manner. (A)** Example images of 2 experimental colonies. Top: colony with pupae in the reproductive phase; bottom: colony with larvae in the brood care phase. The higher dispersion of ants in the colony with larvae indicates foraging activity. Scale bar: 1 cm. **(B)** Example trajectories from 8 individually tagged ants from a single colony with larvae over a 1-hour period. Old and young ants were pharmacologically treated with inotocin or with a control peptide. The remaining 8 ants in the colony (not displayed here) were not treated (S10 Fig). **(C–F)** Average distance traveled by ants in 15-minute time windows over the course of 5 hours following pharmacological treatment ($n = 20$ ants for each treatment from 10 replicate colonies with larvae and 10 colonies with pupae; error bars show the standard error). Nonlinear regression (least squares fit and first- and second-order polynomials). Comparison of curves with extra sum-of-squares F-test. **(C)** In the presence of larvae, old ants treated with inotocin (red) forage more than old ants treated with a control peptide (blue) ($p < 0.0002$, Bonferroni corrected). **(D)** In colonies with pupae, no difference in foraging activity was found between old ants treated with inotocin (red) and the control peptide (blue) ($p > 0.99$, Bonferroni corrected). **(E)** No difference in foraging activity was detected between young ants treated with inotocin (red) and young ants treated with the control peptide (blue) in colonies with larvae ($p = 0.109$, Bonferroni corrected). **(F)** Young ants in colonies with pupae foraged more when treated with inotocin (red) as compared to ants treated with the control peptide (blue) ($p < 0.0002$, Bonferroni corrected). The data underlying this figure can be found in S3 Data and https://doi.org/10.5281/zenodo.4813078.

were precisely controlled (see Materials and methods). Each replicate colony contained 8 young (3 weeks old) and 8 old (4 months old) ants. A total of 10 replicate colonies had larvae, i.e., were in the brood care phase, and 10 replicate colonies had pupae, i.e., were in the reproductive phase (Fig 4A). All young and old ants were derived from single age cohorts from the same large colony to avoid batch effects across experimental replicates. All ants were color tagged for automated individual tracking. In each experimental colony, 2 young ants and 2 old ants each were treated with inotocin and the control peptide. After a 1-hour recovery period, colonies were recorded for 5 hours, and the total distance traveled of each ant was monitored (Fig 4B). Given that tasks in insect societies are spatially organized (foraging, exploring, and waste disposal occur away from the nest, whereas nursing occurs inside the nest), activity-related measures such as spatial location and distance traveled correlate well with foraging and are often used as proxies for foraging activity (e.g., [20,42,69]).

We found that old ants treated with inotocin displayed increased foraging activity in the presence of larvae compared to ants treated with the control peptide ($p < 0.0002$; Fig 4C), while no effect of inotocin treatment was observed when old ants were in the presence of pupae ($p = 0.998$; Fig 4D). This recapitulates the results from our initial pharmacological experiment (S9 Fig). Moreover, it shows that elevated inotocin levels do not simply increase locomotor activity, but do so only in the presence of the stimulating social signal. In other

words, inotocin treatment of old ants specifically amplifies their response to larvae. In contrast, we found no behavioral difference between young ants treated with inotocin and young ants treated with control peptide in colonies with larvae ($p = 0.106$; Fig 4E). However, young ants treated with inotocin in colonies with pupae (i.e., in the reproductive phase) slightly increased their activity compared to young ants treated with the control peptide ($p < 0.0002$; Fig 4F). Unlike in old ants, inotocin treatment in young ants thus appears to specifically modulate the response to pupae.

## Conclusions

Our study shows that inotocin levels in the ant brain naturally increase with age and correlate with the propensity to forage in age-matched individuals. Furthermore, pharmacological manipulations show that increased inotocin levels can indeed increase the time spent outside the nest and the distance traveled by ants, possibly by modulating behavioral response thresholds to specific social cues. We speculate that these cues could be larval pheromones, as inotocin- and inotocin receptor–expressing neurons are located within the SEZ, a center of chemosensory integration in insects. This hypothesis is consistent with studies in mammals, which have shown that oxytocin increases the salience of social cues [70–72]. Taken together, our results suggest that inotocin plays a role in modulating behaviors that correlate with age in ants, such as social foraging. The finding that inotocin can increase foraging activity might seem surprising given that previous work found that knockdown of *inotocin* expression in *Lasius* ants increases locomotor activity while having no effect on social behaviors such as allogrooming, antennation, and trophallaxis [41]. However, that study used a different experimental approach, did not control for age, and focused on metabolic effects. Given our finding that the behavioral response to inotocin treatment is highly contingent on age and social context, we suggest that these previous results and the current study interrogate different behavioral states.

Interestingly, our pharmacological manipulations revealed an unexpected interaction between inotocin treatment, worker age, and social context. While, in accordance with our expectation, inotocin amplified the response to larvae in old workers, in young workers, it specifically affected the behavior in the presence of pupae. In other words, the social context in which elevated inotocin levels increase foraging activity appears to change as a function of worker age. Given that *O. biroi* does not normally forage in the presence of pupae, i.e., in the reproductive phase, our result with young workers is difficult to interpret and should be treated with caution. Additional experiments are required to verify and further dissect this specific finding. However, if our interpretation is correct, it implies that other factors that interact with inotocin must also change as a function of age. One obvious candidate is the inotocin receptor. Indeed, expression of the *inotocin receptor* gene increases with age in *C. fellah* [42]. The receptor is broadly expressed in the head, thorax, and abdomen in *C. fellah*, but, at the cellular level, it had previously only been localized in the oenocytes of the abdominal fat body tissue [42]. Here, we show through in situ hybridization that in *O. biroi*, the inotocin receptor is indeed expressed in the brain and additional peripheral tissues. We performed a reanalysis of published RNA-seq data [52,73] and found no difference in inotocin peptide or receptor expression between the reproductive and brood care phase, suggesting that changes in expression occur over longer timescales (S11 Fig). Additional work is now required to assess whether and how inotocin receptor expression changes in *O. biroi* as ants age. Our discovery of alternative splicing of the inotocin receptor transcript also opens the possibility of age-related changes in receptor splice variants. Such age-dependent changes could constitute a mechanism underlying the age-dependent effects of inotocin treatment observed here. In fact, changes in

functional neuropeptide receptor levels and sites of receptor expression that correlate with behavioral castes, diet, and presence/absence of brood have been shown for the short neuropeptide F receptor (sNPFR) in honeybees [74] and ants [75]. In mammals, the actions of oxytocin and vasopressin depend on their site of release from afferent terminals as well as the presence of receptors in target areas [72]. The site of expression of the receptors is relevant for the regulation of behavior. For example, this mechanism underlies differences in social behavior between polygamous and monogamous voles [76–79] and in the response of mice to pups calls [71].

Oxytocin/vasopressin-related neuropeptides are highly conserved across invertebrates and vertebrates and modulate a range of behaviors, including feeding and food seeking, mating, and offspring provisioning [7,8,71,80]. In particular, these neuropeptides have been implicated in modulating social behavior and social cognition, foremost interactions between adults and their offspring [7,80,81]. As mentioned previously, oxytocin enhances the response to pup calls in the auditory cortex of female mice, enabling pup retrieval [70,71], and, in the nematode *Caenorhabditis elegans*, the oxytocin/vasopressin-related neuropeptide nematocin modulates interactions between adults and larvae via a larval pheromone [82]. Oxytocin/vasopressin-related neuropeptides thus modulate the salience of social cues by acting on the underlying neurocircuitry in a wide range of taxa. Although the relevant signals emanating from *O. biroi* larvae and pupae remain to be identified, our study suggests that inotocin indeed acts on the neural circuits responding to these signals. By modulating the response to immature stages, inotocin thus contributes to behavioral individuality and division of labor in ant societies.

## Materials and methods

### Ant rearing

*O. biroi* colonies were maintained in a climate room at 25°C and 75% relative humidity under constant light. Colonies were kept in airtight plastic containers (10 × 10 cm) for general maintenance, and in airtight Petri dishes (5-cm diameter) for behavioral experiments. Both types of containers had a plaster of Paris floor. Ants were fed 3 times weekly with fire ant (*Solenopsis invicta*) brood, and the containers were cleaned and watered as needed. *O. biroi* clonal line B was used for all experiments in this study, with the exception of ants from clonal line A that were used for the initial description of inotocin staining patterns (Fig 2, S2 Fig). Clonal lines A and B are genetically defined in [43].

### Characterizing the inotocin system in the clonal raider ant

**Inotocin receptor amplification and cloning.** RNA was isolated in TRIzol using the RNeasy (Qiagen, Germantown, Maryland, USA) purification kit with DNAse I (Qiagen) on-column digestion as in [52] from a sample that contained adult ants, larvae, and pupae and reverse-transcribed into cDNA. Full-length cDNAs for the *O. biroi* inotocin receptor were PCR amplified with primers 5′-CCACCATGTCGTACGATTTGAGCTCATCGT-3′ and 5′-TTACC CGAATATTTTTGAGCTCGCAAGTCG-3′, using the Easy-A high fidelity polymerase (Agilent, Santa Clara, California, USA) to generate PCR products with 5′-Kozak consensus sequences and 3′-A overhangs. The inotocin receptor cDNAs were cloned into the pcDNA3.3 TOPO TA cloning vector (Invitrogen, Waltham, Massachusetts, USA) following manufacturer's guidelines.

The full-length *inotocin receptor* cDNA sequence that was cloned and used for calcium assays is ObiroiItr_A (GenBank MW660838; for protein sequence, see S1 Fig). We also tested splice variant ObiroiItr_H (GenBank MW660845; for protein sequence, see S1 Fig), which lacks TM4-7, for calcium activity.

**Cell-based assays for calcium flux imaging.** Human embryonic kidney 293T (HEK293T) cells were cultured in DMEM supplemented with 10% FBS at 37°C in a tissue culture incubator

with a humidified atmosphere containing 5% $CO_2$. Cells were plated in 10-cm dishes, and, when they reached 60% to 70% confluency, they were transfected with a pcDNA3.3 vector containing approximately 1 ng of inotocin receptor cDNA or an empty vector as a control using Lipofectamine 2000. Cells were transfected directly into poly-D-lysine–coated 384-well plates and incubated for 48 hours at 37°C. Subsequently, cells were incubated with FLIPR Calcium 4 Assay kit dye for 1 hour in the incubator. The fluorescence (excitation, 488 nm; emission, 530 nm) of the wells was recorded using a Flex Station II 384 (Molecular Devices, San Jose, California, USA), and a dose curve of inotocin (0 to 3 μM) was applied to the wells, in sextuplicate (see below for inotocin peptide synthesis). We calculated the $EC_{50}$ values using Prism (GraphPad Software version 8) nonlinear regression, 4-parameter logistic curve fit. Dose curves were repeated at least 3 times on independently prepared samples and different days. Results were consistent across replicates, and example dose curves are shown in Fig 1C and S4A Fig. Control experiments are shown in S2–S5 Figs.

**Immunostaining protocol for inotocinergic cells.** A custom goat polyclonal anti-inotocin/neurophysin antibody was designed to recognize an epitope in the neurophysin domain with the sequence CRKIDADLNVMFPGNEASSETFP. The antibody was generated and affinity purified by YenZym (Brisbane, California, USA). Specificity of the antibody response to the immunizing peptide in vitro was assessed with ELISA (by YenZym).

Whole-mount staining of brains was performed following a protocol described in [83]. Briefly, *O. biroi* brains were dissected in cold PBS, pH 7.4. The samples were then fixed by incubation in ice-cold 4% (wt/vol) paraformaldehyde solution in PBS for 3 hours at room temperature. Samples were washed in PBS once, followed by three 20-minute washes in PBS containing 0.5% Triton-X (PBT) at room temperature on a shaker. Samples were blocked in 1% bovine serum albumin (BSA) in PBS for 30 minutes, washed in PBS 0.01% Tween for 5 minutes, and incubated for 24 hours at 4°C with our goat anti-inotocin/neurophysin antibody (1:10,000) containing solution in 1% BSA and 0.5% Triton-X in PBS solution. The next day, samples were washed 3 times for 10 minutes with PBS Tween (0.01%), incubated with a secondary antibody (Alexa Fluor 488 donkey anti-goat solution, 1:250), Alexa Fluor 555, or 647 phalloidin (Thermo Fisher Scientific, Waltham, Massachusetts, USA, 1:50 of 6.6 μM stock solution), and DAPI (1:1,000) in 1% BSA and 0.5% Triton-X in PBS solution for 2 hours, and washed 5 times in PBS. Brains were mounted with Dako mounting medium between 2 cover slips separated by a stack of 2 reinforcement labels (Avery 5720, Avery Products Corporation, Framingham, Massachusetts, USA), mounted on a frosted slide, and sealed using clear nail polish. Brains were imaged using a Zeiss LSM 880 NLO laser scanning confocal microscope (Carl Zeiss, Inc., Thornwood, New York, USA). Images were acquired at a pixel resolution of 1024 × 1024, keeping configuration settings fixed within experiments. The ImageJ open source program FIJI was used to generate 3D projections of the images for display. *D. melanogaster* Canton S flies were dissected, stained, and imaged for inotocin/neurophysin following the same protocol.

**Inotocin and inotocin receptor mRNA fluorescence in situ hybridization chain reaction (HCR) for whole-mount adult brains.** Ant brains were dissected in ice cold 4% paraformaldehyde in RNAse free PBS on ice and fixed in 4% paraformaldehyde in 1x PBS at 4°C for a minimum of 2 hours. Brains were then washed 5 times with 0.5% PBT for 10 minutes each, dehydrated stepwise in EtOH (30%, 50%, 70%, and 100% (3×), 10 minutes each), and incubated overnight in 100% EtOH at 4°C. Brains were then rehydrated stepwise in EtOH (70%, 50%, and 30%, 10 minutes each), incubated in 5% fresh $CH_3COOH$ for 5 minutes at 4°C, washed 5 times with cold PBS (5 minutes each), and fixed in 2% PFA at room temperature for 1 hour. Finally, brains were washed with 0.5% PBT (5×, 10 minutes each), washed in fresh 1% $NaBH_4$ at 4°C (3×, 10 minutes each), and washed with 4°C PBS (5×, 5 minutes each).

We obtained nucleic acid probes (see below) and hairpin amplifier sequences from Molecular Instruments (Los Angeles, California, USA) to detect the neurophysin/inotocin

mRNA producing cells in the brain following an optimized protocol of their in situ HCR protocol for samples in solution [84].

Samples were pre-hybridized in probe hybridization buffer (Molecular Instruments) for 30 minutes at 45˚C, then hybridized with hybridization buffer at 45˚C containing the 6 probes at 1 pmol each and incubated at 45˚C overnight on a thermoblock. They were then washed with 100% preheated wash solution (Molecular Instruments) for 10 minutes, followed by a diluted series of washes in 5x sodium chloride citrate, Tween 20 (5x SSCT) for 15 minutes each (75% wash buffer/25% 5x SSCT, 50% wash buffer/50% 5x SSCT, 25% wash buffer/75% 5xSSCT, 100% 5x SSCT), and a final wash in 100% 5x SSCT for 30 minutes. Samples were then incubated with amplification buffer (Molecular Instruments) for 30 minutes at room temperature. We prepared 30 pmol of each fluorescently labeled hairpin by snap cooling 1 μL of 3 μM stock in hairpin storage buffer (heated to 95˚C for 90 seconds and cooled to room temperature in a dark drawer for 30 minutes) and added 50 μL of amplification buffer at room temperature. The samples were then transferred to hairpin solution and incubated overnight in the dark at room temperature. Finally, samples were washed in 5x SSCT (5×, 10 minutes each) at room temperature, transferred to SlowFade Diamond Antifade mountant, incubated over night at 4˚C, and then mounted.

For double in situ immunostaining, in situ hybridization was performed first, followed by the neurophysin/inotocin immunolabeling protocol described above.

Inotocin HCR probes used (Molecular Instruments):

>P1

gtccaaagcattggcccaaatgattagggccacacgacggacattctcga

>P2

taagatactgctccccacttgcgtcaccgcaaaagcaacaatgatagaat

>P3

cttgcgagcaacaaatcccatttgtagcgcatctagcagtatttccacga

>P4

aatgaactttcgcgatgtcctgaagatactccgtatcacaacgcagcact

>P5

ttacgatcgttgccaacatcggaaattctgcacgatgtgtccatatggca

>P6

tatgtttctgatgttcctatgaaacagccaatactgggaccgcagcagat

Amplifier used: B2 Alexa Fluor 546 (Molecular Instruments).
Inotocin receptor probes used (Molecular Instruments):

>P1

gtactcgtctctcaggtcgtccgacacctccagagacgacggggatgatg

>P2

cagggcaaacaggaccaggccattgccaatgatcgtgatgagaaatatgc

>P3

tgcaggatgaagaagtacatcctggtgaactttcgccgctggtaacatcg

>P4

ttatgtcccaggcgagctgcggcagaacgtcaagcagacccgtcagcagg

>P5

aacgcaactcgtgagactatttagcagactgagaatagtgaaagcagctc

>P6

aattggtcaacgccgcacgtagctccctgttgaatgcaaagtaaatccag

>P7

gaatatttttgagctcgcaagtcgtgagattctggaaataaacgacgatg

Amplifier: B1 Alexa Fluor 647 (Molecular Instruments).

## Comparisons of inotocin levels between different groups

Experimental colonies were composed of age-matched, approximately 1 month old workers, and 3- to 4-day-old larvae in airtight Petri dishes (5-cm in diameter, corresponding to approximately 25 ant body lengths) with a plaster of Paris floor. All workers and larvae were clonally related and sourced from the same stock colony (clonal line B). All workers were harvested from the same cohort and had therefore eclosed within a day of each other, due to the synchronized reproduction of *O. biroi*. From the time they were harvested (1 to 3 days post-eclosion) until the start of the experiment, workers were kept together in a box and allowed to go through a full colony cycle. All workers were tagged with color marks on the thorax and gaster using oil-paint markers (uni Paint Markers PX-20 and PX-21). Experimental colonies contained 16 workers and a matching number of larvae. This 1:1 larvae-to-workers ratio corresponds to the estimated ratio found in a typical (i.e., large and healthy) laboratory stock colony in the brood care phase. *O. biroi* is myrmecophagous, and colonies were fed ad libitum with pupae of fire ant (*S. invicta*) minor workers.

The experiments were conducted in a climate room at 25°C and 75% relative humidity under constant light (*O. biroi* is blind, and its behavior is not affected by light). The colonies were recorded until they entered the reproductive phase, and ants were kept in their colonies until they started the next brood care phase. At the beginning of the next brood care phase, ants that had behaved as foragers or nurses (see below for behavioral analysis) during the previous brood care phase were dissected.

Behavioral data were acquired using an automated scan-sampling approach, in which a picture of each colony was taken at regular intervals by a custom automated tracking setup [20]. This resulted in the acquisition of 8,227 frames per colony collected over 11 days. Individual ants were detected and identified using custom software developed in MATLAB ([20]; software available at https://doi.org/10.5281/zenodo.1211644).

Based on previous assessments [20], the ant identification algorithm on average correctly identifies 77.1% of the ants that can be manually identified in each image (i.e., 22.9% of ants are missed) and assigns the correct identity to 94.4% of the identified ants (i.e., an incorrect identity is assigned 5.6% of the time).

**Behavioral data analysis of age-matched ants.**    We restricted our behavioral analyses to the brood care phase. The spatial distribution of each ant was quantified as the two-

dimensional root–mean–square deviation (r.m.s.d.):

$$r.m.s.d. = \sqrt{\frac{\sum_i \left( (x_i - \bar{x})^2 + (y_i - \bar{y})^2 \right)}{n}},$$

where $(x_i, y_i)$ are the coordinates of the focal ant in frame i, $(\bar{x}, \bar{y})$ are the coordinates of the center of mass of the focal ant's overall spatial distribution in the brood care phase, and $n$ is the number of frames in which the focal ant was detected. r.m.s.d. is bounded between 0 and r, the radius of the Petri dish. Workers that spend a lot of time in the nest with the brood (e.g., nursing the larvae) and little time performing outside tasks (foraging, waste disposal) have low r.m.s.d. values, while workers that spend comparatively more time away from the brood have higher r.m.s.d. values.

In each colony, up to 3 individuals with the highest and lowest r.m.s.d. values that had survived until the end of the experiment were collected for brain dissections and stainings. We only included brains that were intact after the dissection and staining procedures in the final analysis.

**Young and old ants.**   To compare young and old ants, we separated a cohort of approximately 500 callows (newly eclosed ants) from a stock colony and kept them as an age-matched cohort until they were approximately 20 weeks old and at the peak of the brood care phase. From the same stock colony, we separated another cohort of callows that were kept as an age-matched cohort until they were approximately 3 weeks old and likewise were at the peak of the brood care phase. We then dissected all ant brains in parallel for immunohistochemistry analysis.

**Regular workers and intercastes.**   To compare regular workers and intercastes during the brood care phase, we established a colony of callow regular workers and a separate colony of callow intercastes from a single cohort from the same large stock colony. We waited approximately 5 weeks, until the ants had progressed through a reproductive phase and were at the peak of the subsequent brood care phase. We then dissected all ant brains in parallel for immunohistochemistry analysis.

**Image analysis.**   Confocal images were deconvoluted using AutoQuant deconvolution software. We used the IMARIS (Bitplane, Zurich, Switzerland) software to semiautomatically segment the inotocinergic cells and the punctae outside the cell bodies in each brain image stack. We used the total intensity of all fluorescent voxels in each image to quantify inotocin fluorescence intensity. Statistical analyses were performed using Prism (GraphPad Software version 8). An unpaired *t* test (2-tailed) was used to identify significant differences between experimental groups. We used a Bonferroni correction to account for the 3 comparisons in each experimental setup (intensity in cell bodies, number of punctae, and intensity of punctae).

## Pharmacological manipulations

**Inotocin peptide synthesis.**   Mature *O. biroi* inotocin peptide (CLITNCPRG) was synthesized and cyclized at The Rockefeller University Proteomics Resource Center, along with a scrambled version of the peptide (NRPCGLTCI) as a negative control with identical amino acid composition.

**Colony setup.**   For the initial experiment (S9 Fig), we set up 6 experimental colonies of line B ants that were color tagged as previously described. Each colony consisted of 16 ants in an airtight Petri dish with a 5-cm diameter; 8 age-matched 4-week-old and 8 age-matched approximately 20-week-old ants. We added ten 3- to 4-day-old larvae to the colonies. We used Logitech (Lausanne, Switzerland) C910 USB cameras to record 5 hours after the treatments, and we used ImageJ to split the videos in batches of 5,000 frames (corresponding to

18.75-minute intervals). We sampled every 10th frame from the 5,000 frames and overlaid the resulting 500 frames into a single image using ImageJ. In each image, we manually counted the number of instances of ants that were outside the nest as a proxy for foraging behavior. Color tagging was used to differentiate inotocin treated from mock treated and untreated ants. The nest was defined as a region of high density of ants in the 500 image projections.

For the full experiment (Fig 3), we set up 20 experimental colonies of line B ants that were color painted as previously described. Each colony consisted of 16 ants in an airtight Petri dish with 5-cm diameter. Eight ants in each colony were 3 weeks old, and 8 ants in each colony were 20 weeks old. We added ten 3- to 4-day-old larvae to each of 10 colonies and 10 white (young) pupae to each of the remaining 10 colonies. Brood items often die soon after introduction to a colony, so we added 2 extra brood items per colony to ensure there were 8 alive at the start of the experiment. After the ants had settled for 3 days, we adjusted the number of brood items to 8 per colony, after which we performed the pharmacological treatments. The rationale of having a 1:2 larvae-to-workers ratio in this experiment was to assess the possible effect of increased inotocin levels on worker behavior without saturating the brood signal, as ant larvae regulate worker foraging behavior in a dose-dependent manner [50].

On the day of the experiment, colonies were not fed. To avoid disturbance to the colony as much as possible, half the colony was left untreated, and half was treated. The treatments consisted of immersing the ants in 1 mM solution of freshly dissolved inotocin or control peptide diluted in water and kept on ice. The immersion took place in 1.5-ml Eppendorf tubes that were gently tapped against the bench top for 7 consecutive minutes, after which the liquid was removed using a 1.5-ml transfer pipette. The remaining liquid was removed with a Kimwipe, and ants were transferred to their source colony. In each colony, we treated 2 ants per age category with inotocin and 2 ants per age category with the control peptide. Ants were allowed to recover for 1 hour after the treatment before the start of the video recordings.

**Behavioral tracking.** Videos were recorded for 5 hours at 5 frames per second using Logitech C910 USB cameras. Individual ants were tracked using anTraX [85]. In short, ants were segmented using a background subtraction and fixed threshold procedure. Blobs in consecutive frames were threaded into sequences using the Horn–Schunk optical flow method and then classified as single ants or multiple ants based on blob size and linkage criteria. Single ant blob sequences were classified into their appropriate IDs as detailed in [85]. Ants that were detected as single blobs were deemed "active," and their location was recorded (Fig 3B). The activity level of an ant was defined as the total distance traveled, and an activity time course was created for each ant by summing its activity in time bins of 15 minutes for the entire duration of the experiment. All ants in a colony were tracked. The activity across ants from a given condition from all experimental colonies was averaged in each time bin. The resulting curves were fitted with first- or second-order polynomials using nonlinear regression (least squares fit). The fitted curves were compared using extra sum-of-squares F-tests to select for the best fitted model and to compare between fitted curves, and statistical significance thresholds were adjusted for 2 comparisons within experiments using Bonferroni corrections (old inotocin versus old mock and young inotocin versus young mock). Raw video files for behavioral data can be found at https://doi.org/10.5281/zenodo.4813078.

## Supporting information

**S1 Fig. The *inotocin receptor* gene produces 10 splice variants in *O. biroi*.** Alignment of the inotocin receptor splice variants inferred from RNA-seq data [52], along with the *Lasius niger* inotocin receptor as a reference (from [56]). The transmembrane domains required for proper receptor function are highlighted in red. The data underlying this figure can be found in S4

Data. RNA-seq, RNA sequencing.
(TIF)

**S2 Fig. The inotocin receptor is functional when expressed in HEK293T cells. (A)** The ItcR responds in a dose-dependent manner to inotocin. **(B)** No response is detected when stimulated with the vehicle alone (buffer). **(C)** No $Ca^{2+}$ influx is detected when ItcR expressing cells are stimulated with a scrambled control peptide. **(D)** No $Ca^{2+}$ influx is detected when cells transfected with an empty vector are stimulated with inotocin (100 nM), and cells transfected with an empty vector are healthy and show calcium responses when stimulated with ATP (10 μM). The traces for the inotocin response when stimulating the inotocin receptor are the same data displayed in each panel. The data underlying this figure can be found in S5 Data. HEK293T, human embryonic kidney 293T.
(TIF)

**S3 Fig. The inotocin receptor does not require co-transfection with a G-protein when expressed in HEK293T cells to elicit calcium responses when stimulated with inotocin. (A and B)** *itcR* co-transfected with empty vector (no G protein) show calcium responses when stimulated with inotocin. **(C and D)** Co-transfection with Gq alpha 16. **(E and F)** Co-transfection with Gq alpha with Gi alpha C term chimera. **(G and H)** *itcR* co-transfection with Gq alpha with Go alpha C term chimera. In **A**, **C**, **E**, and **G** cells were grown at 37˚C for 24 hours and then transferred to 28˚C for an additional 16 to 24 hours. In **B**, **D**, **F**, and **H** cells where constantly grown at 37˚C for the same amount of total time. The data underlying this figure can be found in S6 Data. HEK293T, human embryonic kidney 293T.
(TIF)

**S4 Fig. Human oxytocin and vasopressin peptides also elicit calcium responses in cells expresing the *Ooceraea biroi* inotocin receptor. (A)** Inotocin $Ca^{2+}$ response in cells transfected with *itcR* (independent experiment from the one displayed in Fig 1C). **(B)** Oxytocin $Ca^{2+}$ response in cells transfected with *itcR*. **(C)** AVP $Ca^{2+}$ response in cells transfected with *itcR*. Experiment shown in **A** is one of 3 replicates, and each concentration of inotocin was assayed in sextuplicate in all replicates. Experiments in **B** and **C** were performed once and include triplicate measurments. Previous experiments have shown that HEK293T cells do not endogenously express the oxytocin or vasopressin receptors (Supporting information references [1–5] in S1 Text). The data underlying this figure can be found in S7 Data. See S1 Text for Supporting information methods. AVP, arginine vasopressin; HEK293T, human embryonic kidney 293T.
(TIF)

**S5 Fig. Calcium responses of HEK293T cells that express the inotocin receptor. Lack of response in cells transfected with an empty vector or with a truncated receptor isoform.** Each image is one frame extracted from recordings of HEK293T cells stimulated with inotocin. The panel on the left is from before stimulation, and the panel on the right is from after the introduction of inotocin. **(A and B)** Calcium signal is detected (red) in cells expressing the inotocin receptor (ObiroiItr_A, for protein sequence see S1 Fig) and G protein 16. **(C and D)** No calcium signal is detected when stimulating the cells transfected with an empty vector. **(E and F)** No calcium signal is detected when a truncated splice variant of the receptor is expressed (ObiroiItr_H, which lacks TM4-7, for protein sequence see S1 Fig). See S1 Text for Supporting information methods. HEK293T, human embryonic kidney 293T.
(TIF)

**S6 Fig. Inotocin antibody staining controls. (A)** Control staining without primary antibody. **(B)** A *Drosophila melanogaster* brain stained with the inotocin antibody (green). *Drosophila melanogaster* lacks the inotocin system, and, accordingly, no antibody staining is visible. Nuclei are stained with DAPI (blue), and actin is stained with phalloidin (magenta). Scale bar in each panel represents 100 μm. AMMC, antennal mechanosensory and motor center; MB, mushroom body; SEZ, subesophageal zone.
(TIF)

**S7 Fig. Behavioral data for age-matched foragers and nurses.** Ten colonies of 16 ants each were monitored using automated behavioral tracking, and ants were ranked according to their r.m.s.d. values (see Materials and methods). Individuals used for inotocin measurements (Fig 3A) are shown in red (nurses) and blue (foragers). We were not able to always include the most extreme workers and the same number of workers from all colonies, because some brains were lost or damaged during dissection, staining, or imaging. The data underlying this figure can be found in S8 Data. r.m.s.d., root–mean–square deviation.
(TIF)

**S8 Fig. Comparisons of inotocin levels between different types of ants. (A** and **B)** No differences were found between age-matched nurses and foragers ($n \geq 12$; unpaired $t$ test, Bonferroni corrected). **(C)** Older ants have more inotocin punctae than young ants ($n \geq 10$; unpaired $t$ test, $^*p = 0.045$, Bonferroni corrected). **(D)** Old ants show more inotocin staining (a proxy for the amount of inotocin) than young ants ($n = 10$; unpaired $t$ test, $^{***}p < 0.0003$, Bonferroni corrected). **(E** and **F)** No differences were found between age-matched intercastes and regular workers ($n \geq 9$; unpaired $t$ test, Bonferroni corrected). The data underlying this figure can be found in S2 Data.
(TIF)

**S9 Fig. Experimentally elevating inotocin levels leads to increased foraging in mixed-age colonies.** Total number of instances that ants treated with inotocin (red) or a control peptide (blue) were found outside the nest (a proxy for foraging activity) in a projection image of 500 frames corresponding to 18.75-minute windows over the course of 5 hours. Error bars show the standard error. In colonies composed of adult ants of varying ages with larvae, ants foraged more when treated with inotocin than when treated with the control peptide. An average of all ants per treatment per time point was graphed, and a second degree polynomial curve was fitted via nonlinear regression using PRISM GraphPad. The fitted curves were compared using the extra sum-of-squares F-test ($n = 6$; $p = 0.0002$). The data underlying this figure can be found in S9 Data.
(TIF)

**S10 Fig. Elevated inotocin levels lead to increased foraging in an age- and social context dependent manner.** The data here are also depicted in Fig 4, with the inclusion here of the results for untreated ants (gray) that were part of the colony. Average distance traveled by ants in 15-minute time windows over the course of 5 hours (proxy for foraging activity) following pharmacological treatment ($n = 20$ ants for each treatment from 10 replicate colonies with larvae and 10 colonies with pupae; error bars show the standard error). **(A** and **B)** In the presence of larvae, old ants treated with inotocin (red) had higher average distance traveled than ants under any other experimental condition. **(C** and **D)** In colonies with pupae, only young ants treated with inotocin (red) increased their average distance traveled compared to ants under any other experimental condition. Control treated (blue) and non-treated (gray). Non-treated ants show relatively higher activity early in the experiment because experimental ants are still recovering from the immersion treatment. The data underlying this figure can be found in S3

Data and https://doi.org/10.5281/zenodo.4813078.
(TIF)

**S11 Fig. No differences in** *inotocin* **nor** *inotocin receptor* **mRNA expression in brains were found between clonal raider ants in the brood care phase (colonies with larvae) compared with the reproductive phase (colonies with pupae).** Gene expression data for *inotocin* (**A** and **B**) and *itcR* (**C** and **D**) were analyzed from the control samples originally used in [52,73]. Data from [52] for **A** and **C** and data from [73] for **B** and **D**. Samples in each case consist of age-matched workers in the brood care phase with larvae or age-matched workers in the reproductive phase with pupae. Wald tests were performed on each group to calculate significance. Gene expression units represent variance-stabilized transformed read counts. The data underlying this figure can be found in S10 Data and in https://doi.org/10.5281/zenodo.4562942. See S1 Text for Supporting information methods.
(TIF)

**S1 Data. Supporting data for Fig 1.**
(XLSX)

**S2 Data. Supporting data for Fig 3 and S8 Fig.**
(XLSX)

**S3 Data. Supporting data for Fig 4 and S10 Fig.**
(XLSX)

**S4 Data. Supporting data for S1 Fig.**
(RTF)

**S5 Data. Supporting data for S2 Fig.**
(XLSX)

**S6 Data. Supporting data for S3 Fig.**
(XLSX)

**S7 Data. Supporting data for S4 Fig.**
(XLSX)

**S8 Data. Supporting data for S7 Fig.**
(XLSX)

**S9 Data. Supporting data for S9 Fig.**
(XLSX)

**S10 Data. Supporting data for S11 Fig.**
(XLSX)

**S1 Text. Supporting information methods and Supporting information references.**
(DOCX)

## Acknowledgments

We thank the members of the Kronauer, Bargmann, Sakmar, Ruta, Vosshall, Maimon and Steller laboratories at The Rockefeller University for reagents, access to laboratory equipment, and input throughout the project. We thank Thomas Sakmar for valuable advice on the calcium imaging experiments. We thank Zachary Coto and Aylesse Sordillo for assistance with preliminary experiments. We thank Stephany Valdés Rodríguez, Ni-Chen Chang, and Amelia

Ritger for technical assistance. We are also thankful to Nipun Basrur who provided advice on in situ hybridization. We thank Henry Zebroski III from the Rockefeller University Proteomics Resource Center for peptide synthesis and Alison North, Christina Pyrgaki, and Tao Tong at the Rockefeller Bio-Imaging Resource Center for confocal imaging advice. I.F.-P. is also thankful to Ian A. E. Butler for providing feedback on various versions of the manuscript and to Alma Piñeyro-Nelson for providing office space and support during the writing of the manuscript at the Centro de Ciencias de la Complejidad, Universidad Nacional Autónoma de México. I.F.-P. is a Global Consortium for Reproductive Longevity and Equality Junior Scholar. This is Clonal Raider Ant Project paper #15.

## Author Contributions

**Conceptualization:** Ingrid Fetter-Pruneda, Jennifer L. Garrison, Cornelia I. Bargmann, Daniel J. C. Kronauer.

**Data curation:** Ingrid Fetter-Pruneda, Yuko Ulrich, Asaf Gal.

**Formal analysis:** Ingrid Fetter-Pruneda, Taylor Hart, Yuko Ulrich, Asaf Gal.

**Funding acquisition:** Ingrid Fetter-Pruneda, Cornelia I. Bargmann, Daniel J. C. Kronauer.

**Investigation:** Ingrid Fetter-Pruneda, Taylor Hart, Yuko Ulrich, Asaf Gal, Peter R. Oxley, Leonora Olivos-Cisneros, Margaret S. Ebert.

**Methodology:** Ingrid Fetter-Pruneda, Manija A. Kazmi, Jennifer L. Garrison.

**Project administration:** Ingrid Fetter-Pruneda, Daniel J. C. Kronauer.

**Resources:** Cornelia I. Bargmann, Daniel J. C. Kronauer.

**Software:** Yuko Ulrich, Asaf Gal.

**Supervision:** Daniel J. C. Kronauer.

**Validation:** Ingrid Fetter-Pruneda.

**Visualization:** Ingrid Fetter-Pruneda.

**Writing – original draft:** Ingrid Fetter-Pruneda, Daniel J. C. Kronauer.

**Writing – review & editing:** Ingrid Fetter-Pruneda, Taylor Hart, Yuko Ulrich, Asaf Gal, Peter R. Oxley, Leonora Olivos-Cisneros, Margaret S. Ebert, Manija A. Kazmi, Jennifer L. Garrison, Cornelia I. Bargmann, Daniel J. C. Kronauer.

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
