## [Editor Report · Decision Letter 0]

15 Jun 2020

Dear Dr Fetter Pruneda, 

Thank you for submitting your manuscript entitled "An oxytocin/vasopressin-related neuropeptide modulates social foraging behavior in ants" for consideration as a Short Reports by PLOS Biology.

Your manuscript has now been evaluated by the PLOS Biology editorial staff, as well as by an academic editor with relevant expertise, and I'm writing to let you know that we would like to send your submission out for external peer review.

Please re-submit your manuscript within two working days, i.e. by Jun 17 2020 11:59PM.

Kind regards,

Roli Roberts

Senior Editor

PLOS Biology

---

## [Decision Letter · Decision Letter 1]

3 Aug 2020

Dear Dr Fetter Pruneda,

Thank you very much for submitting your manuscript "An oxytocin/vasopressin-related neuropeptide modulates social foraging behavior in ants" for consideration as a Short Report at PLOS Biology. Your manuscript has been evaluated by the PLOS Biology editors, an Academic Editor with relevant expertise, and by three independent reviewers.

You will see that while two of the reviewers are largely positive about your study, reviewer #1 questions the magnitude of the advance over related studies such as Liutkeviciute & Gruber 2018, and raises several significant concerns which should be addressed. The other two reviewers also have several requests, and all three reviewers would like to see inotocin receptor expression patterns.

In light of the reviews (below), we will not be able to accept the current version of the manuscript, but we would welcome re-submission of a much-revised version that takes into account the reviewers' comments. We cannot make any decision about publication until we have seen the revised manuscript and your response to the reviewers' comments. Your revised manuscript is also likely to be sent for further evaluation by the reviewers.

We expect to receive your revised manuscript within 3 months. 

**IMPORTANT - SUBMITTING YOUR REVISION**

*Re-submission Checklist*

*Published Peer Review*

*PLOS Data Policy*

*Blot and Gel Data Policy*

Sincerely,

Roli Roberts

Senior Editor,

rroberts@plos.org,

PLOS Biology

REVIEWERS' COMMENTS:

Reviewer #1:

#### Summary

The author investigate regulation of social behavior by inotocin in the model ant system, O. biroi. First, they clone a full length isoform of the receptor and shows that it responds to the synthetic peptide when expressed in 293 cells. Then they perform IF with a custom antibody and FISH to identify two neurons that produce inotocin in the ant brain and their projections. Next, they show that levels of inotocin peptide (by IF) are higher in foragers and older ants. Finally, they use a pharmacological approach to show increased "foraging" in old ants in presence of larvae and young ants in presence of pupae.

#### Overall critique

The idea of studying inotocin in ants is interesting because of the important social roles of its orthologs in various animals and because, as the authors point out, Drosophila lacks this pathway. However, the current study adds little to our knowledge about this system. Figures 1-3 show results similar to those previously reported in other ant species and Fig. 4 show a confusing set of results using an assay that does not measure foraging behavior. Overall the study is preliminary and lacks novelty; therefore, I do not recommend its publication.

#### Major points 

1. The abstract claims that the experimental evidence in this study suggest that "inotocin plays an important role in mediating age-polyethism". This is a big overstatement. The experiments show a correlation between age, foraging, and inotocin expression and no causal link with age polyethism.

2. Fig. 1C: missing controls, what about another neuropeptide receptor or at least the empty control mentioned in the methods? Also why not use the control shuffled peptide also mentioned in the methods?

3. Fig. 1-3: the first three figures recapitulate the data shown in Liutkeviciute & Gruber FASEB 2018 Fig. 1, 3, and 4. Although it is of some value to show that those conclusions hold in this different ant system, the novelty of these data is quite limited. I acknowledge that the authors detect protein levels rather than mRNA but that seems a small detail in this context.

4. Fig. 3C: I applaud the authors for showing this piece of evidence which does not fit well with the overall idea of inotocin regulating foraging but it does raise some important questions about the conclusions and the proposed model. What if they were to treat with inotocin the intercaste?

5. Fig. 4 and S5: even though manipulation of the inotocin pathway in ants has been reported before (Liutkeviciute & Gruber 2018) this is the first gain-of-function experiment to my knowledge and therefore contains new information. Even so, this experiment is not complete and cannot be used to draw the conclusions that the authors wish to draw. 

5.1. The treatment appears to be immersion of the ants in a inotocin-containing solution. Although creative, this seems highly unusual and should be openly mentioned in the description of the results rather than only in the material and methods. Further, it remains to be demonstrated that inotocin actually gets inside the ants, that it acts as an agonist in vivo at these concentrations, and that it acts in the brain. To the latter point, Liutkeviciute & Gruber noted that the inotocin receptor is broadly expressed and by delivering the peptide immersion it can be safely concluded that all receptors in the body will be activated.

5.2. The assay seems to measure general locomotion and not "foraging". In fact an effect of inotocin on locomotion was already reported by Liutkeviciute & Gruber. In my view, this cannot be considered a regulation of "social behavior".

5.3. The results are hard to interpret. Inotocin only has an effect on old ants with larvae and young ants with pupae? Even the authors seem unsure of what this means as "O biroi does ot normally forage in the presence o pupae". If inotocin is normally already high in old workers but low in young workers, wouldn't one expect a stronger effect in the latter? Are there differences in the receptor?

5.4. Finally, the different "social contexts" are only introduced in this very last experiment. Why not also in the analysis of inotocin expression in Fig. 3? And waht about receptor expression? Is it possible that different social contexts regulate receptor expression rather than inotocin itself?

#### Minor point

- why are the legends interspersed with the text even if the figures are at the end?

- Fig. 1B: the legend states that the transmembrane domains are "required for receptor function". This needs experimental support or a citation. If the authors are correct it seems that 8/10 of the isoforms would be nonfunctional. 

- Fig. 4B: it is unclear if these example tracks are from colonies with larvae, pupae etc.

Reviewer #2:

This thorough study of inotocin in the Ooceraea clonal raider ant demonstrates that inotocin signaling is likely to be a conserved feature of ant biology with the potential to regulate responsiveness to social cues and division of labor. The methodology and experimental design are impressive and appear robust, including the commendable use of assays of peptide levels as well as treatments with the inotocin peptide in combination with the tracking of individual ant movements in different social contexts. The results of the study build on and recapitulate findings from two recent studies conducted in ants from a different subfamily. Among the key findings here are that inotocin stimulates foraging behavior (more movement throughout an arena) and that this effect interacts with age and social stimulus (particularly the presence of larvae that need feeding). Among the open questions are where the inotocin receptor localizes in Ooceraea and whether the localization is consistent with its function in the brain and/or with its function in desiccation resistance observed in Camponotus ants, which would provide very interesting insight into questions of evolutionary lability and pleiotropy of inotocin signaling. Such insight would undoubtedly raise the impact of the current study, which I think would be the main question in whether it rises to the level of PLOS Biology, but the findings are impressive in the current form. In reading the manuscript and the other two studies from the formicine ants I learned some interesting things and would consider this time well spent for anyone interested in the proximate underpinnings of the social lives of ants. I would simply suggest that the title should be revised - if "in ants" is changed to "in an ant" or the title is otherwise softened it may more appropriately characterize the incomplete nature of the picture of the generalities of the roles of inotocin signaling in ants at present. 

Reviewer #3:

The manuscript entitled "Oxytocin/vasopressin-related peptide modulates foraging in ants" by Fetter-Pruneda et al. describes a role for the neuropeptide inotocin in regulating division of labor in colonies of the clonal raider ant. While multiple different papers have demonstrated associations between several signaling pathways and worker's division labor in multiple social insect species, this well-written manuscript is one of very few that actually demonstrate a possible causal link between a conserved peptidergic pathway and the neural regulation of a derived social behavioral trait. Below are more specific comments that I hope the authors will find useful: 

1. It is possible that regulation of the inotocin receptor plays as important role as the ligand in regulating DOL in this species. Since the gene encoding it has been identified, and the authors can generate beautiful ISH data, it could be very informative to examine is expression as well. At minimum, the authors should discuss in more depth what is known about the regulation of peptidergic systems, and should probably include a discussion of how the regulation of peptide secretion and receptor signaling could play a role in regulating behavior independent of quantitative changes in ligand expression.

2. Demonstrating that the candidate inotocin receptor show response to the ligand in physiologically-relevant concentrations is an important aspect of this study. However, since the authors have used human HEK293 cells to express an insect GPCR, it is somewhat surprising that they were able to observe a response without the co-transfection of a promiscuous Gqα subunit. Adding cAMP activation data could be one way to further convince readers that the measured intracellular signals are not somehow artifactual. Adding images showing HEK cells at baseline and post-activation would further increase confidence in the approach. 

3. Is it possible that inotocin injections induce general hyperactivity, which might be interpreted as increased foraging? It seems it would be possible to extract the rates of individual activity from the tracking data the authors already have.

4. Adding high-res images of stained brains from young nurses and old foragers will allow readers to appreciate the observed differences qualitatively. 

5. It's not clear if the authors tracked injected animals only or all colony members. If all then they should also show the tracking data for untreated animals. For example, it would be very interesting to know if increased foraging in older injected animals increased the activity in the non-focal older ants.

---

## [Decision Letter · Decision Letter 2]

16 Apr 2021

Dear Dr Fetter Pruneda,

Thank you for submitting your revised Short Report entitled "An oxytocin/vasopressin-related neuropeptide modulates social foraging behavior in the clonal raider ant" for publication in PLOS Biology. I've now obtained advice from two of the original reviewers and have discussed their comments with the Academic Editor. 

Based on the reviews, we will probably accept this manuscript for publication, provided you satisfactorily address the remaining points raised by the reviewers. Please also make sure to address the following data and other policy-related requests.

IMPORTANT:

a) Please attend to the remaining concerns from reviewer #1; these involve an addition control and some quantification for two of your new Supplementary Figures.

b) We note that you now have 5 main Figures. The maximum allowable for a Short Report is 4. Please could you combine two of these Figures into a single Figure?

c) Please attend to my Data Policy requests further down. Essentially you will need to supply the numerical values that underlie Figs 1C, 4ABC, 5BCDEF, S1 (alignment), S2ABCD, S3ABCDEFGH, 4ABC, S7, S8ABCDEF, S9, S10ABCD, S11ABCD, and cite the location of the data in the respective Figure legends.

We expect to receive your revised manuscript within two weeks. 

*Published Peer Review History*

*Early Version*

Sincerely,

Roli Roberts

Senior Editor,

rroberts@plos.org,

PLOS Biology

DATA POLICY:

Regardless of the method selected, please ensure that you provide the individual numerical values that underlie the summary data displayed in the following figure panels as they are essential for readers to assess your analysis and to reproduce it: Figs 1C, 4ABC, 5BCDEF, S1 (alignment), S2ABCD, S3ABCDEFGH, 4ABC, S7, S8ABCDEF, S9, S10ABCD, S11ABCD. NOTE: the numerical data provided should include all replicates AND the way in which the plotted mean and errors were derived (it should not present only the mean/average values).

DATA NOT SHOWN?

REVIEWERS' COMMENTS:

Reviewer #1:

This revised manuscript contains some necessary control for the in vitro assay, additional data on the localization and levels of receptor expression and some important clarifications and other text changes. Overall I find the manuscript improved. Although I remain somewhat skeptical of the authors interpretation of the key experiment (now in Fig. 5) as measuring "foraging" and "social behavior", I appreciate their point of view and, considering also the enthusiasm expressed by the other reviewers, I join them in supporting publication of this study. I have only a couple of minor points.

- Lines 47-8: I don't think the authors demonstrated that "Inotocin signaling [...] contributes to behavioral individuality and division of

labor in ant societies." I suggest to tone this last sentence down.

- The authors added important controls for Fig 1C in the supplement, but they also present new data in Fig. S4 that lack proper controls. Are the authors claiming that the O. biroi itcR binds and respond to human vasopressin and oxytocin? If so an empty vector control must be shown to exclude that the 293 cells used respond naturally to these neuropeptides.

- Fig.S5: possibly due to my own lack of familiarity with this technique I cant' interpret these microscopy images. Why not show a quantification as in Fig. 1C, Fig. S3, 4?

Reviewer #3:

The authors did an excellent job revising this manuscript, including the addition of new data and analyses. The revised manuscript provides important insights into both the evolution and mechanisms for the regulation of behavior in social insects. I have no additional concerns.

---

## [Editor Report · Decision Letter 3]

3 Jun 2021

Dear Ingrid,

On behalf of my colleagues and the Academic Editor, Lars Chittka, I'm pleased to say that we can in principle offer to publish your Short Reports "An oxytocin/vasopressin-related neuropeptide modulates social foraging behavior in the clonal raider ant" in PLOS Biology, provided you address any remaining formatting and reporting issues. These will be detailed in an email that will follow this letter and that you will usually receive within 2-3 business days, during which time no action is required from you. Please note that we will not be able to formally accept your manuscript and schedule it for publication until you have made the required changes.

PRESS: We frequently collaborate with press offices. If your institution or institutions have a press office, please notify them about your upcoming paper at this point, to enable them to help maximise its impact. If the press office is planning to promote your findings, we would be grateful if they could coordinate with biologypress@plos.org. If you have not yet opted out of the early version process, we ask that you notify us immediately of any press plans so that we may do so on your behalf.

Thank you again for supporting Open Access publishing. We look forward to publishing your paper in PLOS Biology. 

Sincerely,

Roli Roberts

Roland G Roberts, PhD 

Senior Editor 

PLOS Biology